# A Lymphoid Organ Specific Anti-Lipopolysaccharide Factor from *Litopenaeus vannamei* Exhibits Strong Antimicrobial Activities

**DOI:** 10.3390/md19050250

**Published:** 2021-04-28

**Authors:** Mingzhe Sun, Shihao Li, Xinjia Lv, Jianhai Xiang, Yuanan Lu, Fuhua Li

**Affiliations:** 1Key Laboratory of Experimental Marine Biology, Institute of Oceanology, Chinese Academy of Sciences, Qingdao 266071, China; mzhsun@qdio.ac.cn (M.S.); lvxinjia@qdio.ac.cn (X.L.); jhxiang@qdio.ac.cn (J.X.); 2Laboratory for Marine Biology and Biotechnology, Qingdao National Laboratory for Marine Science and Technology, Qingdao 266237, China; 3Center for Ocean Mega-Science, Chinese Academy of Sciences, Qingdao 266071, China; 4Environmental Health Laboratory, Department of Public Health Sciences, University of Hawaii at Manoa, Honolulu, HI 96822, USA; yuanan@hawaii.edu; 5The Innovation of Seed Design, Chinese Academy of Sciences, Wuhan 430072, China

**Keywords:** anti-lipopolysaccharide factor, LPS-binding domain, lymphoid organ, white spot syndrome virus, *Vibro parahaemolyticus*

## Abstract

Different shrimp species are known to possess apparent distinct resistance to different pathogens in aquaculture. However, the molecular mechanism underlying this finding still remains unknown. One kind of important antimicrobial peptides, anti-lipopolysaccharide factors (ALF), exhibit broad-spectrum antimicrobial activities. Here, we reported a newly identified ALF from the shrimp *Litopenaeus vannamei* and compared the immune function with its counterpart in the shrimp *Fenneropenaeus chinensis*. The ALF, designated as LvALF8, was specifically expressed in the lymphoid organ of *L. vannamei*. The expression level of *LvALF8* was apparently changed after white spot syndrome virus (WSSV) or *Vibrio parahaemolyticus* challenges. The synthetic LBD peptide of LvALF8 (LvALF8-LBD) showed strong antibacterial activities against most tested Gram-negative and Gram-positive bacteria. LvALF8-LBD could also inhibit the in vivo propagation of WSSV similar as FcALF8-LBD, the LBD of LvALF8 counterpart in *F. chinensis*. However, LvALF8-LBD and FcALF8-LBD exhibited apparently different antibacterial activity against *V. parahaemolyticus*, the main pathogen causing acute hepatopancreatic necrosis disease (AHPND) of affected shrimp. A structural analysis showed that the positive net charge and amphipathicity characteristics of LvALF8-LBD peptide were speculated as two important components for its enhanced antimicrobial activity compared to those of FcALF8-LBD. These new findings may not only provide some evidence to explain the distinct disease resistance among different shrimp species, but also lay out new research ground for the testing and development of LBD-originated antimicrobial peptides to control of shrimp diseases.

## 1. Introduction

Shrimp products are known to be a significant source of aquatic animal protein [1] and shrimp aquaculture production reached more than 3,000,000 metric tons in 2018, according to the statistics of the Food and Agriculture Organization (FAO) (http://www.fao.org/fishery/statistics/en, accessed on 7 February 2021). Among the cultured shrimp species, Pacific whiteleg shrimp *Litopenaeus vannamei* accounts for the main part of the global shrimp production [2]. Since the introduction into China in the 1990s [3], *L. vannamei* has rapidly replaced its close relative *Fenneropenaeus chinensis* and become the main cultured shrimp species [1]. One of the main reasons for the replacement is that *L. vannamei* appears to be more tolerant to the hypersaline environment and pathogens infection than *F. chinensis* [4,5,6]. However, investigation on the molecular mechanism underlying the difference is very limited.

Shrimps mainly rely on the innate immune response to control and clear invading pathogens following infection, and antimicrobial peptides (AMPs) are the key components of their innate immune system [7]. Anti-lipopolysaccharide factor (ALF), crustin, penaeidin, and stylicin are the main types of AMPs identified in shrimp [7]. Previous studies have revealed that ALFs, with a common feature of the functional domain termed the LPS-binding domain (LBD) and a disulfide loop formed by two conserved cysteine residues, possess a broad-spectrum antimicrobial activity against Gram-positive bacteria, Gram-negative bacteria, fungi and viruses [8,9]. Knock-down of ALF genes by double-strand RNA (dsRNA) could significantly increase the mortality rates of shrimp *Marsupenaeus japonicus* infected by *Vibrio* spp. [10] and result in the rapid proliferation of endogenous bacteria in *Exopalaemon carinicauda* [11]. The recombinant FcALF5 protein could directly interact with the envelope protein VP24 of white spot syndrome virus (WSSV) and inhibit its infection to shrimp after pre-incubation with the virus [12]. These data revealed that ALFs are essential effectors in innate immune system of shrimp. Therefore, we speculated that different antimicrobial activities of AMPs, such as ALFs, might be related to the discrepancy of disease resistance among different shrimp species.

Present studies on shrimp have shown that different ALF family members co-existed in one organism [13,14,15], and different ALF family members from a single organism showed varied tissue expressions and distinct antimicrobial activity [16,17,18,19]. Eight ALF family members have been reported from *F. chinensis* [20,21]. The synthetic LBD peptides from diverse FcALFs showed distinct antimicrobial activities against various bacteria and WSSV [17,20]. Application of the LBD peptides of FcALF1, FcALF2, FcALF5, and FcALF7 effectively inhibited WSSV propagation in vivo [17]. The LBD peptides of FcALF1, FcALF2, FcALF7, and FcALF8 could effectively inhibit the growth of Gram-positive bacteria *Micrococcus luteus*, while FcALF1, FcALF4, Fc ALF6, FcALF7 and FcALF8 showed antibacterial activities on Gram-negative bacteria *Escherichia coli* [17,20]. Several ALF genes have been identified in *L. vannamei* and a few were functionally characterized. It was reported that the transcription and protein levels of LvAV-K were up-regulated following WSSV or *Vibrio anguillarum* infection [22]. Single nucleotide polymorphism (SNP) of the LvALF1 gene was related to WSSV-resistance [23]. Until now, there have been no reports about the comparison on the activity of ALFs from different shrimp species, which might provide some explanations on their different disease resistance.

In our previous study, lymphoid organ of *F. chinensis* was identified to specifically express *FcALF8* and its synthetic LBD peptide exhibited high inhibition activity against bacterial pathogens, both in vivo and in vitro [20]. In this study, we report the identification of *LvALF8*, the homologue of *FcALF8*, from the lymphoid organ of the Pacific whiteleg shrimp *L. vannamei*. The antibacterial and antiviral activities of LvALF8 were characterized, including comparative analysis between LvALF8 and FcALF8. These new findings might provide new insights into the function of AMPs in crustaceans and lay out creative instructions for the design of LBD-originated antimicrobial drugs for shrimp aquaculture.

## 2. Results

### 2.1. Sequences and Phylogenetic Analysis of LvALF8

The full-length cDNA of *LvALF8* was 600 bp, with an open reading frame of 399 bp encoding 132 deduced amino acid (aa) residues. A signal peptide with 27 aa was predicted. The putative LPS-binding domain (LBD) located between Cys61 and Cys82 with two conserved cysteine residues (Figure 1).

The full-length sequences of LvALF8 and other 39 ALF proteins excluding signal peptides from 12 crustacean species (Table 1) were used to construct the neighbor-joining phylogenetic tree, to assess the evolutionary relationships among them. As shown in Figure 2, all mature ALFs could be classified into seven categories, among which the bootstrap replications are <40%. LvALF8 had a closer evolutionary relationship with FcALF8.

### 2.2. Tissue Distribution of LvALF8 Transcripts

Fifteen tissues, including brain, epidermis, eyestalk, gill, hemocytes, hepatopancreas, heart, intestine, muscle, lymphoid organ (Oka), ovary, stomach, testis, thoracic ganglia, and ventral nerve cord were collected for the expression analysis of *LvALF8* transcripts. The results showed that *LvALF8* was specifically expressed in the lymphoid organ (Figure 3).

### 2.3. Expression Profiles of LvALF8 after WSSV or Vibrio Parahaemolyticus Infection

The expression profiles of *LvALF8* in Oka after viral and bacterial infection were measured by qRT-PCR. After WSSV infection, the expression level of *LvALF8* significantly decreased at 6 h post infection (hpi), and then increased at 24 hpi (Figure 4A). However, expression pattern of *LvALF8* after *V. parahaemolyticus* infections very different and showed an initially significant increase at 3 hpi (the highest) and 6 hpi, but dramatically decreased at 12 hpi and 24 hpi (Figure 4B).

### 2.4. Anti-Bacterial Activities of LvALF8-LBD Peptide

The anti-bacterial activities of LvALF8-LBD were analyzed by detecting the minimal inhibitory concentration of synthetic LvALF8-LBD peptide on different bacteria. As shown in Table 2, LvALF8-LBD showed strong anti-bacterial activities against most bacteria with different concentrations, including 1–2 µM against *V. parahaemolyticus*, *V. harveyi*, *V. alginolyticus*, and *V. owensii*, 2–4 µM against *Photobacterium damselae* and *Staphylococcus epidermidis*, 4–8 µM against *E. coli*, *Kurthia gibsonii*, and *S. aureus*.

### 2.5. In Vivo Anti-Bacterial Function of LvALF8-LBD and FcALF8-LBD

Simultaneous injection of two peptides of LBD8 together with *V.*
*parahaemolyticus* could inhibit the in vivo proliferation of the bacteria. In the LvALF8-LBD + *V. parahaemolyticus* (designated as LV) group, the total viable bacteria are 4.62 × 10^2^ cfu/g in hepatopancreas, which were significantly lower than the 1.36 × 10^3^ cfu/g detected in the hepatopancreas of shrimp group (designated as FV) injected with the FcALF8-LBD + *V. parahaemolyticus*. In addition, the total bacterial counts in the hepatopancreas of these two treated groups were both lower than the count (1.42 × 10^4^ cfu/g) detected in the hepatopancreas of the GFP + *V. parahaemolyticus* (designated as GV) group (Figure 5A). Further analysis shrimp hepatopancreas for bacteria identification showed that the predominant bacteria were *V. parahaemolyticus* in the GV group (green spots in Figure 5C). However, *V. parahaemolyticus* were undetectable in the LV group and less detected in the FV group (2.82 × 10^2^ cfu/g), which was significantly lower than that in the GV group (8.82 × 10^3^ cfu/g) (Figure 5B). Moreover, the numbers of other bacteria such as *V. owensii* and *V. harveyi* detected in the LV group were less compared to the control group (yellow spots in Figure 5C).

### 2.6. Agglutination Activity of LvALF8-LBD and FcALF8-LBD

Agglutination test on the two peptides showed that both LvALF8-LBD and FcALF8-LBD could lead to distinct agglutination of the bacteria *V. parahaemolyticus* in comparison with the negative controls (PBS and GFP groups) (Figure 6).

### 2.7. In Vivo Anti-Virus Function of LvALF8-LBD and FcALF8-LBD

The copy numbers of WSSV were measured and compared for different groups at 48 h post injection. The WSSV copy numbers in LvALF8-LBD + WSSV (designated as LW) and FcALF8-LBD + WSSV (designated as FW) groups were significantly reduced compared to the GFP + WSSV (designated as GW) group (Figure 7). However, no significant difference in WSSV copy number was detected between LW and FW groups.

### 2.8. Structure Analysis of LvALF8-LBD and FcALF8-LBD

Sequence alignments of LvALF8-LBD and FcALF8-LBD were performed to understand the relationship between the amino acid sequence of LBD and its high antibacterial activity. The result showed that 18 of the 22 residues in LBD were the same between LvALF8-LBD and FcALF8-LBD. The residues #10, 13, 18 and 20 of FcALF8-LBD were replaced from isoleucine, glutamine, threonine, and isoleucine to leucine, arginine, serine, and valine in LvALF8-LBD, respectively (Figure 8A). The physicochemical properties and amphipathic characters of LvALF8-LBD and FcALF8-LBD were analyzed by ProtParam tool and HeliQuest analysis. The helix wheel diagram (Figure 8B) showed that the positively charged hydrophilic amino acid residues of LvALF8-LBD were located on two sides, whereas the hydrophobic residues were on the other two sides, presenting a perfect amphipathic structure for LvALF8-LBD, which is similar to that of FcALF8-LBD. The analysis also showed that the arginine residue in LvALF8-LBD peptide replaced the glutamine residue of the FcALF8-LBD peptide, making basic residues enriched on one side. The physiochemical parameters of LvALF8-LBD were shown in Table 3. The positive net charge and hydrophilicity of LvALF8-LBD were higher than those of FcALF8-LBD peptide.

## 3. Discussion

Like many other animals, shrimp are subjective to bacterial infections, including various forms of *Vibrio* pathogens (*V. alginolyticus*, *V. harveyi*, and *V. parahaemolyticus*) and *P. damselae* [24,25,26,27]. Antimicrobial peptides play essential roles in host defense against these pathogens. ALF is one of the well-studied antimicrobial peptides in crustaceans; eight different ALF genes have recently been identified in *F. chinensis* [20]. Importantly, the lymphoid organ specific *FcLAF8* encoded by one of the ALF genes exhibited strong activities against several *Vibrio* pathogens, including *V. alginolyticus*, *V. harveyi*, and *P. damselae* [20]. Here, we have identified a new ALF, *LvALF8*, from the whiteleg shrimp *L. vannamei*. *LvALF8* and *FcALF8* are homologues existing in the two kinds of shrimps, sharing high sequence identity, closer evolutionary relationship and the same tissue distribution. Therefore, they provided a good case to study whether antimicrobial peptides could contribute to the distinct pathogen resistance abilities of different shrimp species.

As one typical antimicrobial peptide, LvALF8 exhibits strong antimicrobial activities against both bacteria and viruses. *LvALF8* showed significant transcriptional responses in the lymphoid organ upon WSSV or *V. parahaemolyticus* infection, indicating that it might play important immune roles during pathogen infections. Although we only compared the expression of *LvALF8* in its mainly expressed organ after infection, it is possible that the expression of *LvALF8* may significantly increase in other organs. LvALF8 had strong in vitro and in vivo activities against several *Vibrio* pathogens. It had better inhibition against *V. alginolyticus* and *V. harveyi* than several reported ALFs, like LsALF-D1, LsALF-B1 and PmALF-B1 [28]. Moreover, LvALF8 also exhibited stronger in vitro antibacterial activity against Gram-positive pathogens, including *S. aureus* and *S. epidermidis*, as compared to some other AMPs [20,29]. In addition, LvALF8-LBD peptide could lead to the detectable agglutination of *V. parahaemolyticus*, which is consistent to the finding from a modified LBD peptide against several bacteria, including *E. coli* and *V. harveyi* [30]. Agglutination is known to be one of the important host responses to recognize the foreign pathogens, thereby mediating further immune responses in the invertebrate immune defense system [31]. The data showed that the LBD peptide of LvALF8 could not only directly kill pathogens, but also inhibit their infection through modulating host immune responses.

It is shown in this study that both LvALF8 and FcALF8 exhibited strong inhibition against several important pathogenic *Vibrio* pathogens and a similar antiviral activity against WSSV. However, their antibacterial activity against *V. parahaemolyticus* was quite different. *V. parahaemolyticus* is considered as the major shrimp pathogen causing the acute hepatopancreatic necrosis disease (AHPND), an emerging and severe disease of cultured shrimp in recent years [27]. Although both LvALF8-LBD and FcALF8-LBD could agglutinate *V. parahaemolyticus* in vitro, both in vivo and in vitro tests revealed that LvALF8-LBD exhibited much stronger antibacterial activity against *V. parahaemolyticus* as compared to FcALF8-LBD [20]. In addition, much stronger inhibition against *E. coli* and *S. epidermidis* was also detected by using LvALF8-LBD peptide than FcALF8-LBD [20]. These data suggested that one homologous gene encoding an antimicrobial peptide in different shrimp had distinct antimicrobial activities, which might contribute to the distinct pathogen resistance of different shrimp species.

A previous study showed that the distinct functions of AMPs were mainly determined by their amino acid sequences [7]. This study showed that the four residues between LvALF8-LBD and FcALF8-LBD are different from each other. Importantly, the 13th residue, glutamine of FcALF8-LBD, was replaced by a basic amino acid Arginine in LvALF8-LBD. Net cationic charge, amphipathicity and hydrophobicity are considered as the key characteristics of active AMPs [32]. Most ALF belongs to cationic AMPs, in which the basic amino acids acted as an important factor to bind to negatively charged surface in lipid membrane of bacteria [33]. Dathe et al. reported that the net charge and basic amino acids in LBD peptides greatly contribute to their activities, and the loss of basic amino acids in LBD peptides leads to the decrease of antimicrobial functions in *F. chinensis* [34]. Increasing the net charge from magainin 2 could increase its antimicrobial activity [35]. Similarly, decreasing the net charge of AMPs could reduce their antimicrobial activity [36]. No acidic amino acids and a total of six basic amino acids, including three lysine residues and three arginine residues exist in the LBD peptide of LvALF8, resulting in one more net charge than the FcALF8-LBD peptide. The increased net charge of LvALF8-LBD peptide resulting from the one more arginine residue might be responsible for the enhanced antibacterial activity, in comparison with the FcALF8-LBD peptide.

## 4. Materials and Methods

### 4.1. Experimental Animals, Pathogen Stimulation and Tissue Collection

Healthy adult Pacific whiteleg shrimp with a body weight of 2.5 ± 0.4 g, were routinely cultured in our laboratory and used for tissue distribution analysis and pathogen stimulation experiments. All the animal tests were conducted in accordance with the guidelines of the respective Animal Research and Ethics Committees of the Chinese Academy of Sciences. Before any experimental test, shrimp were acclimated in air-pumped circulating sea water at 25 ± 1 °C and fed with commercial food pellet for about a week. Shrimp were selected randomly in each experimental test. Hemocytes, brain, epidermis, eyestalk, gill, hepatopancreas, heart, intestine, muscle, lymphoid organ, nerve cord, stomach, ovary and testis were collected from nine shrimps for tissue distribution analysis and same tissue from three individuals were polled together as one sample.

For the pathogen stimulation experiment, WSSVs were prepared according to the method described by Sun et al. [37], and *V. parahaemolyticus* isolated in our lab were cultured to the mid-logarithmic-phase. Experimental animals were randomly divided into three groups, including PBS control group, WSSV group and V.p group, with 45 shrimps per group. The WSSV was diluted with PBS buffer (137 mmol/L NaCl, 2.7 mmol/L KCl, 10 mmol/L Na_2_HPO_4_, 1.8 mmol/L KH_2_PO_4_, pH 7.4) to the final concentration of 800 copies/μL, and bacteria were diluted with PBS buffer to the final concentration of 10^6^ cfu/mL. Ten microliters of pathogen solution were injected into each shrimp at the III and IV abdominal segments for the WSSV and V.p groups. The equal volume of PBS was injected into each shrimp in the PBS group. Lymphoid organs from nine shrimps were sampled at each of five time points as 0, 3, 6, 12, 24 hpi; one sample for three shrimp and three samples for each time point.

### 4.2. Total RNA Extraction and cDNA Synthesis

The total RNA was extracted by TRIzol reagent (Takara, Kyoto, Japan) and its quality was confirmed by 1% agarose gel electrophoresis. cDNA template was synthesized using 1 µg of total RNA for each sample using RevertAid First Strand cDNA synthesis Kit (Thermo Fisher Scientific, Waltham, MA, USA), with random primers according to the manufacturer’s protocols.

### 4.3. Sequence Analysis of LvALF8 Gene

The open reading frame (ORF) of *LvALF8* was amplified by PCR with the primers *LvALF8*-F (5′-TCTCGGCAACACGACAGCAAC-3′) and *LvALF8*-R (5′-ACGTCCTTGCAACCGGTCAAT-3′) designed based on the sequence with accession number MF135544 and confirmed by Sanger sequencing. The BLAST algorithm (NCBI, http://www.ncbi.nlm.nih.gov/BLAST/, accessed on 27 October 2020) was used to analyze the nucleotide sequence and deduced amino acid sequence of *LvALF8*. CBS prediction servers (http://www.cbs.dtu.dk/services, accessed on 27 October 2020) were used to predict the signal peptide. Unlike the well characterized and designated ALF family members in *F. chinensis*, the designation in *L. vannamei* is confused. Different ALF protein sequences (Table 1) obtained from the NCBI database were used to construct a phylogenic tree using MEGA-X software (https://www.megasoftware.net/, accessed on 27 October 2020) by the neighbor joining distance algorithm.

### 4.4. Quantitative Real-Time qPCR

The mRNA expressions of *LvALF8* in different tissues were detected by quantitative real-time qPCR with a pair of primers *LvALF8*-qF (5′-TGACGAATCTGCGAACTCCA-3′) and *LvALF8*-qR (5′-CGCCATCTTTGACCAGGGAA-3′) using THUNDERBIRD™ SYBR^®^ qPCR Mix Without ROX (Toyobo, Osaka, Japan). The 18S rRNA (accession number EU920969) amplified via a pair of primers 18S-qF (5′-TATACGCTAGTGGAGCTGGAA-3′) and 18S-qR (5′-GGGGAGGTAGTGACGAAAAAT-3′) was employed as an internal control for cDNA normalization. The PCR product was denatured to produce a melting curve to check the specificity of the PCR product. The expression levels of all selected genes were detected in three biological replicates.

### 4.5. Synthesis of LvALF8-LBD

The LBD peptide of LvALF8, named as LvALF8-LBD, was purchased from a Chinese commercial company (Sangon Biotech, Shanghai, China). The amino acid residue of LvALF8-LBD, Ac-(CSYSTRPYFLRWRLKFKSKVWC)-NH_2_, was amidated in the C-terminal and acetylated in the N-terminal. The two cysteine residues formed a disulfide bond. The synthetic LvALF8-LBD peptide was tested by mass spectrometry and high-performance liquid chromatography. The LBD peptide of FcALF8-LBD, Ac-(CSYSTRPYFIRWQLKFKTKIWC)-NH_2_, was synthesized for comparison analysis. A partial peptide of Green Fluorescent Protein (GFP) (Accession number: AAN41637), Ac-(TTGKLPVPWPTLVTTFSYGVQCFS)-NH_2_, was included as a negative control.

### 4.6. Minimal Inhibitory Concentration (MIC) Assay

The antimicrobial activity of peptides was determined using the minimal growth inhibition concentration (MIC) assay against Gram-positive and Gram-negative bacteria, as described previously [20]. A total of nine different bacterial strains, including six Gram-negative bacteria, *E. coli*, *V. alginolyticus*, *V. harveyi*, *P. damselae*, *V.*
*parahaemolyticus*, and *V. owensii* and three Gram-positive bacteria, *S. aureus*, *S. epidermidis*, and *K. gibsonii*, were used for detection. Briefly, bacterial cells harvested at the mid-logarithmic phase were diluted to 1 × 10^5^ cfu/mL in PBS buffer (137 mmol/L NaCl, 2.7 mmol/L KCl, 10 mmol/L Na_2_HPO_4_, 1.8 mmol/L KH_2_PO_4_, pH 7.4). In this test, 15 µL/well of diluted peptide solutions (1/2-fold serial dilutions with the PBS) were added to a 96-well plate (Corning, New York, NY, USA). The final concentration of peptide in the medium ranged from 64 µM, 32 μM, 16 μM, 8 μM, 4 μM, 2 μM, 1 μM to 0.5 μM. Fifteen microliters PBS and 15 µL pGFP solution were also used as blank group and negative control. Each well of the plate was added 133 µL growth medium, and the mixtures were incubated for 6 to 8 h depending on different bacterial strains. Absorbance at 600 nm for Gram-positive bacteria or at 560 nm for Gram-negative bacteria was determined using a precision micro-plate reader (TECAN infinite M200 PRO, TECAN, Salzburg, Austria). The assay was conducted in triplicate.

### 4.7. Peptide Injection and Pathogens Infection

Considering that FcALF8-LBD did not show strong antibacterial activity to *V. parahaemolyticus* in our previous study [19], the shrimps with body weight of 1.25 ± 0.17 g were used to further analyze the function of LvALF8-LBD and FcALF8-LBD against *V. parahaemolyticus* was through the in vivo analysis of the anti-bacterial activity of two synthetic LBD8 peptides. In this test, the experimental animals were randomly divided into four groups: PBS, GFP + *V. parahaemolyticus* (designated as GV), FcALF8-LBD + *V. parahaemolyticus* (designated as FV), LvALF8-LBD + *V. parahaemolyticus* (designated as LV), and each group contained 30 individuals. For FV and LV groups, *V. parahaemolyticus* at the final concentration of 10^6^ cfu/mL was mixed with 64 μM of FcALF8-LBD or LvALF8-LBD peptides respectively, then 10 μL solution was immediately injected into each shrimp. For the GV and PBS group, each shrimp was injected with 10 μL (10^4^ cfu) of *V. parahaemolyticus* mixed with 64 μM GFP peptide or 10 μL PBS solution alone as negative controls. At 24 h after injection, the hepatopancreas, the target tissue of *V. parahaemolyticus* [38], was collected from 15 sacrificed individuals for each group and pooled at three per sample and five samples per group.

To analyze the antiviral activity of LvALF8-LBD, the shrimps with a body weight of 1.56 ± 0.23 g were used, and these experimental animals were randomly divided into three groups: GFP + WSSV (designated as GW), FcALF8-LBD + WSSV (designated as FW), LvALF8-LBD + WSSV (designated as LW), and 30 animals per group were used. For shrimp in FW and LW groups, 500 copies/μL of WSSV was incubated with 64 μM of FcALF8-LBD or LvALF8-LBD peptides for 2 h at room temperature, respectively, then 10 μL mixture/shrimp was injected for 30 shrimps. For the GW group, each shrimp was injected with 10 μL of WSSV (5000 copies) WSSV pre-incubated with 64 μM GFP peptide for 2 h at room temperature as control. At 48 h after injection, the pleopods were collected from 15 shrimps in each group and pooled at three per sample and five samples per group.

### 4.8. Bacterial Count and Strain Identification

The collected hepatopancreas samples were weighted, crushed, and blended in sterile PBS separately. Then, 100 μL of the suspension with different dilution was seeded onto TCBS agar media. After incubation at 28 °C for 18 h, the number of total bacterial colonies on the plate was counted. Ten single colonies of the dominant bacteria were picked and used as DNA templates for PCR amplification of a partial sequence of the 16S rDNA, with the universal primers 27F and 1492R. Bacterial identification was carried out through sequencing and analyzing the amplified products.

### 4.9. DNA Extraction and WSSV Load Quantification

Total DNA was extracted from pooled pleopods using the Genomic DNA Kit (Tiangen, Beijing, China) with the addition of Protease K (Roche, Mannheim, Germany) at a final concentration of 5.7 mg/mL for digestion, according to the manufacturer’s instructions. Quantification of extracted DNA was conducted by NanoDrop 2000 (Thermo Fisher Scientific, Waltham, MA, USA). SYBR green-based quantitative real-time PCR (qRT-PCR) was used to quantitatively analyze the viral loads in the pleopods according to the method described by Sun et al. [37]. Briefly, The DNA encoding the WSSV envelope protein VP28 was amplified and cloned into pMD19-T simple vector (Takara, Kyoto, Japan). The standard curve was generated using the purified and quantified plasmid. Primers VP28-qF (5′-AAACCTCCGCATTCCTGTGA-3′) and VP28-qR (5′-TCCGCATCTTCTTCCTTCAT-3′) were used to detect the viral loads in the pleopods. Each assay was carried out in quadruplicate.

### 4.10. Bacterial Agglutination Experiment

The bacteria at their logarithmic growth phase were harvested and fixed with a 0.1 mg/mL FITC solution. Then, the FITC-labeled bacteria cells were diluted to 10^7^ cfu/mL, mixed with 64 μM of either FcALF8-LBD peptide solution or LvALF8-LBD peptide solution. The PBS and 64 μM pGFP peptide solution were also used as negative controls. Following 1 hr incubation at room temperature within 1 h, the affected cells were added to the glass slide and observed under an optical Nikon TS100 microscope (Nikon Corporation, Tokyo, Japan).

### 4.11. Structure Analysis on the Peptide LvALF8-LBD

The online program ProtParam tool (http://web.expasy.org/protparam/, accessed on 4 February 2021) and Antimicrobial Peptide Database (http://aps.unmc.edu/AP/main.php, accessed on 4 February 2021) were used to calculate the physicochemical properties of peptides, including net charge and hydrophobic percentage. The online program HeliQuest analysis (http://heliquest.ipmc.cnrs.fr/cgi-bin/ComputParams.py, accessed on 4 February 2021) was used to predict the amphipathic characters of these peptides.

### 4.12. Statistical Analysis

All the data were expressed in the form of mean ± S.E. and analyzed with one-way analysis of variance (ANOVA) and *t*-test comparisons. Differences between treatments and controls were considered significant at *p* < 0.05 and extremely significant at *p* < 0.01.

## 5. Conclusions

In conclusion, a novel ALF gene, *LvALF8*, was detected to be specifically expressed in the lymphoid organ of whiteleg shrimp *L. vannamei*. The LBD peptide of LvALF8 showed a strong antibacterial effect against all the bacteria tested in this study, including six Gram-negative and three Gram-positive bacteria. The LBD peptide of LvALF8 also led to the distinct agglutination with *V. parahaemolyticus*. In vivo tests confirmed its antibacterial activity to *V. parahaemolyticus*. In addition, the peptide exhibited significant antiviral activity against WSSV in shrimp. Both in vivo and in vitro tests showed that LvALF8-LBD peptide-mediated antibacterial activity was much stronger than that detected with FcALF8-LBD, especially to *V. parahaemolyticus*. This might be related to the higher net cationic charge and enriched hydrophobicity on one side, which resulted from a replaced basic amino acid residue in LvALF8-LBD peptide. The present findings may provide new insights into ALF functions in crustaceans and will provide useful information that is essential for the design of LBD-originated antimicrobial substances.

## Figures and Tables

**Figure 1 marinedrugs-19-00250-f001:**
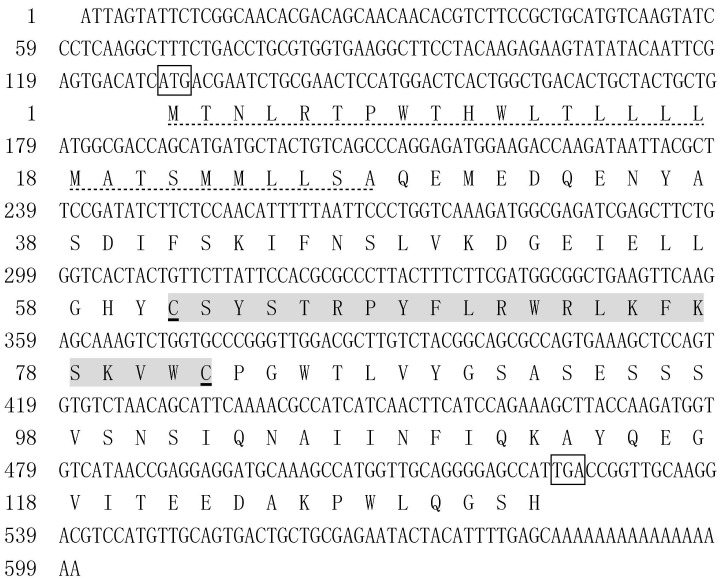
Nucleotide sequence and deduced amino acid sequence of *LvALF8*. The start codon and stop codon were boxed. The dotted underlined residues represented predicted signal peptide. The residues marked with dark background were putative LPS-binding domain and conserved cysteine residues were underlined.

**Figure 2 marinedrugs-19-00250-f002:**
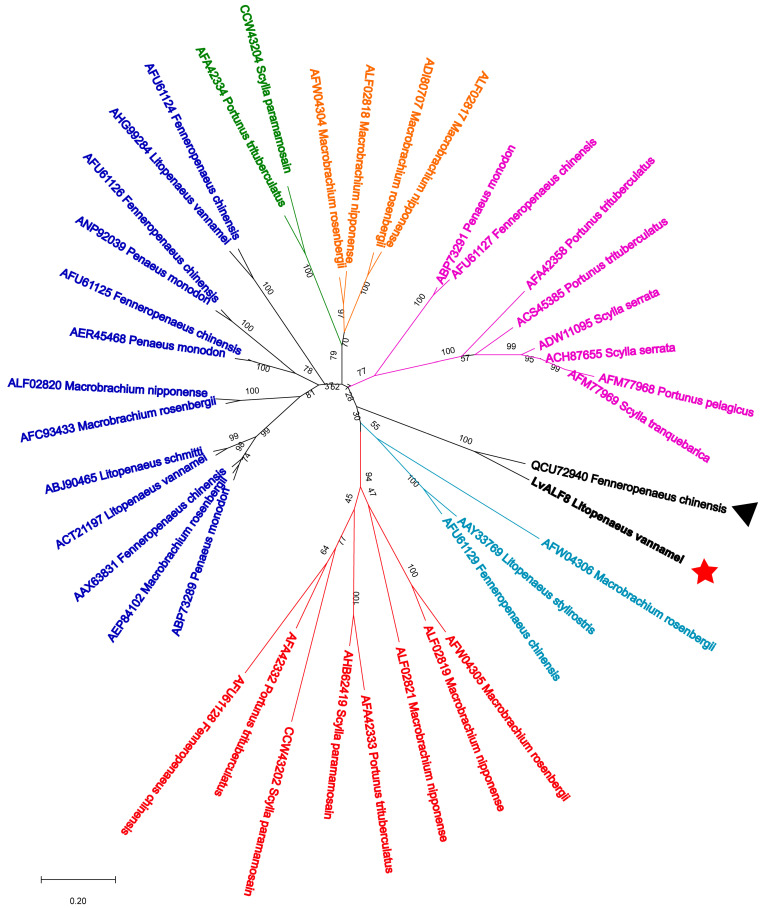
Phylogenetic analyses of anti-lipopolysaccharide factors (ALFs) from twelve crustacean species. The GenBank accession numbers of ALFs were listed in Table 1. Whole deduced amino acid sequences of mature ALFs were used for phylogenic analysis by Neighbor Joining, with the Bootstrap value at 1000. Divergence distance and percentage of bootstrap replications were shown in the figure. Different type ALFs (with <40% of bootstrap replications) were classified into seven branches shown with different colors. LvALF8 sequence was marked with a red five-pointed star (
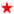
) and FcALF8 sequence was marked with a black triangle (▲).

**Figure 3 marinedrugs-19-00250-f003:**
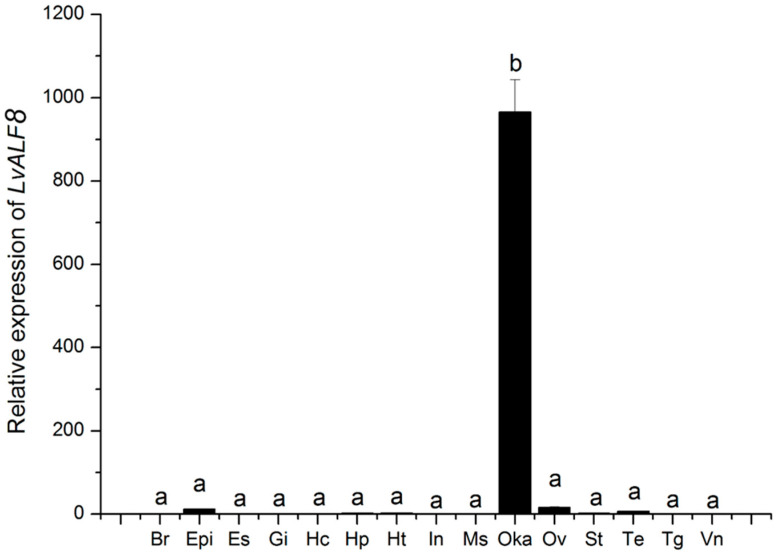
The tissue distribution of *LvALF8* transcripts. Vertical bars represented mean ± S.E. (n = 3). Br, brain; Epi, epidermis; Es, eyestalk; Gi, gill; Hc, hemocytes; Hp, hepatopancreas; Ht, heart; In, intestine; Ms, muscle; Oka, lymphoid organ; Ov, ovary; St, stomach; Te, testis; Tg, thoracic ganglia; Vn, ventral nerve cord. Letters, ‘‘a’’, and “b’’ represented significant differences among tissues at *p* < 0.01.

**Figure 4 marinedrugs-19-00250-f004:**
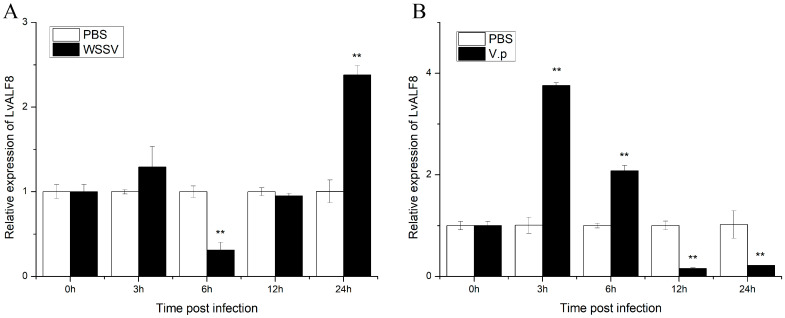
Expression levels of *LvALF8* in the lymphoid organ of shrimp at different time post-WSSV (**A**) or *V. parahaemolyticus* (**B**) challenge. PBS stands for shrimps injected with phosphate-buffered saline (PBS), (WSSV) stands for shrimps injected with white spot syndrome virus (WSSV) and V.p stands for shrimps injected with *V. parahaemolyticus*. Gene expression levels with significant differences (*p* < 0.01) between the two treatments were shown with stars (**).

**Figure 5 marinedrugs-19-00250-f005:**
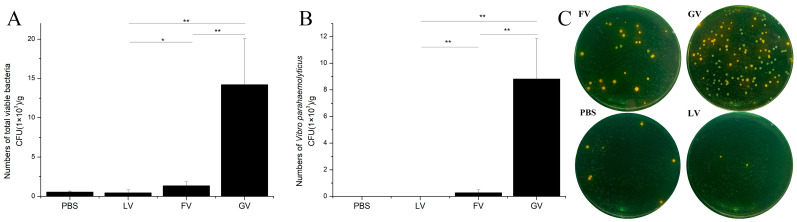
In vivo anti-bacterial function of *LvALF8* in *L. vannamei.* (**A**) Total viable bacteria count in hepatopancreas (Hp) of shrimp injected with synthetic peptide and *V. parahaemolyticus*. (**B**) *V. parahaemolyticus* counts in hepatopancreas (Hp) of shrimp injected with synthetic peptide and *V. parahaemolyticus*. (**C**) Spread plates of hepatopancreas homogenate of shrimp injected with synthetic peptide and *V. parahaemolyticus*. PBS, injected with PBS solution; LV, injected with synthetic LvALF8-LBD peptide and *V. parahaemolyticus*; FV, injected with synthetic FcALF8-LBD peptide and *V. parahaemolyticus*; GV, injected with synthetic GFP peptide and *V. parahaemolyticus*. Bacteria numbers with significant differences between the two treatments were shown with stars (* at *p* < 0.05 and ** at *p* < 0.01).

**Figure 6 marinedrugs-19-00250-f006:**
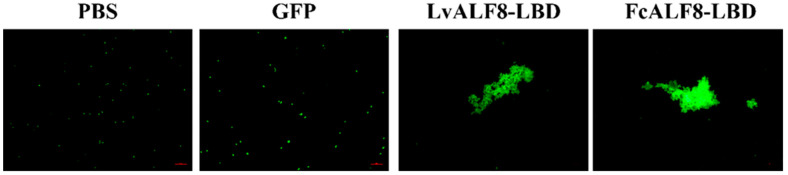
Bacterial agglutination assay of LvALF8. 10^7^ cfu/mL *V. Parahaemolyticus* is incubated with 64 μM LvALF8-LBD, FcALF8-LBD and GFP peptides within 1 h, respectively, and observed under a microscope. Scale bar is 20 μm.

**Figure 7 marinedrugs-19-00250-f007:**
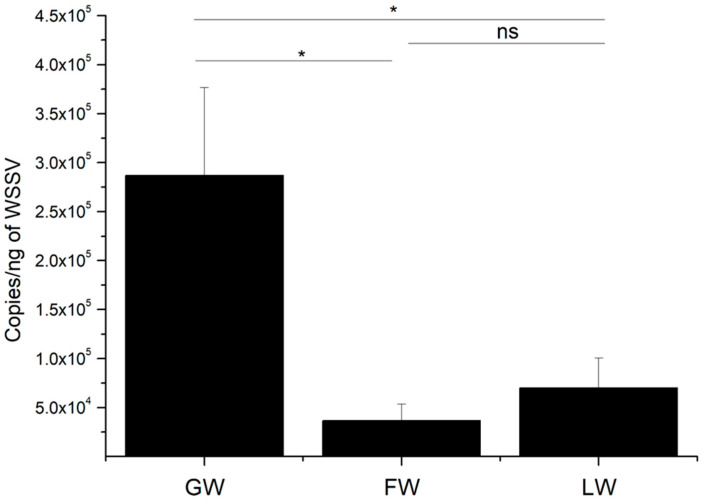
In vivo anti-viral function of LvALF8 in *L. vannamei* after infection of WSSV pre-incubated with peptides. The copy number of WSSV was shown by measuring the content of VP28 per ng DNA. GW, FW and LW showed three different groups injected with WSSV pre-incubated with GFP, FcALF8-LBD and LvALF8-LBD peptides, respectively. Viral copy numbers with significant differences (*p* < 0.05) between the two treatments were shown with star (*). No significant difference was shown with “ns”.

**Figure 8 marinedrugs-19-00250-f008:**
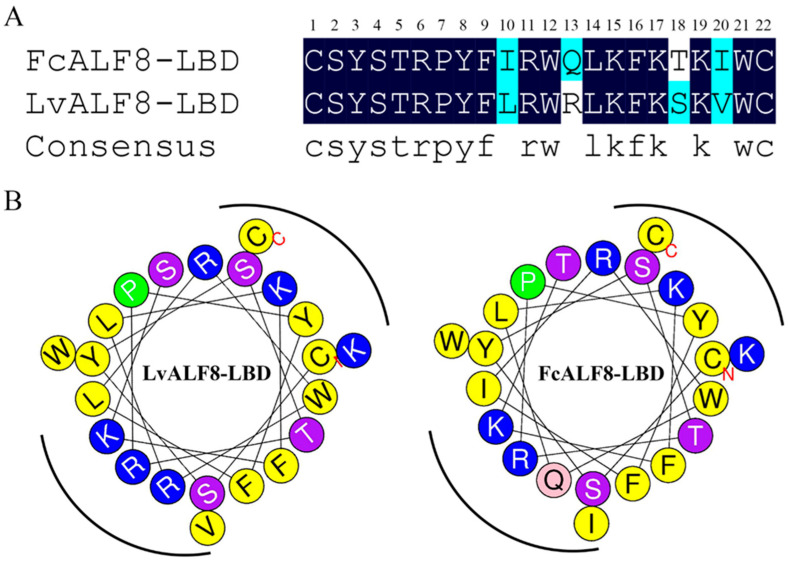
Sequence alignments and helical wheel prediction of LvALF8-LBD and FcALF8-LBD. (**A**) Sequence alignments of LvALF8-LBD and FcALF8-LBD. The 22 amino acid residues of LBD sequences were listed, and their identical residues were marked darkly. (**B**) Helical wheel prediction of the distribution of amino acid side chains: LvALF8-LBD and FcALF8-LBD. The basic residues were shown in blue, the hydrophobic residues were shown in yellow, and the hydrophilic residues with different hydrophilic abilities were shown in other colors.

**Table 1 marinedrugs-19-00250-t001:** The information of ALFs used in the present study.

Species	Gene Name	Accession Number
*Fenneropenaeus chinensis*	Antilipopolysaccharide factor isoform 1	AFU61124
*Fenneropenaeus chinensis*	Antilipopolysaccharide factor isoform 2	AFU61125
*Fenneropenaeus chinensis*	Antilipopolysaccharide factor isoform 3	AFU61126
*Fenneropenaeus chinensis*	Antilipopolysaccharide factor isoform 4	AFU61127
*Fenneropenaeus chinensis*	Antilipopolysaccharide factor isoform 5	AFU61128
*Fenneropenaeus chinensis*	Antilipopolysaccharide factor isoform 6	AFU61129
*Fenneropenaeus chinensis*	Antimicrobial peptide (FcALF7)	AAX63831
*Fenneropenaeus chinensis*	Antilipopolysaccharide factor isoform 8	MH998632
*Macrobrachium nipponense*	Anti-lipopolysaccharide factor 5	ALF02821
*Macrobrachium nipponense*	Anti-lipopolysaccharide factor 4	ALF02820
*Macrobrachium nipponense*	Anti-lipopolysaccharide factor 3	ALF02819
*Macrobrachium nipponense*	Anti-lipopolysaccharide factor 2	ALF02817
*Macrobrachium nipponense*	Anti-lipopolysaccharide factor 1	ALF02818
*Portunus trituberculatus*	Anti-lipopolysaccharide factor isoform 7	AFA42358
*Portunus trituberculatus*	Anti-lipopolysaccharide factor isoform 6	AFA42334
*Portunus trituberculatus*	Anti-lipopolysaccharide factor isoform 5	AFA42332
*Portunus trituberculatus*	Anti-lipopolysaccharide factor isoform 4	AFA42333
*Portunus trituberculatus*	Anti-lipopolysaccharide factor isoform 3	ACS45385
*Portunus pelagicus*	Anti-lipopolysaccharide factor precursor	AFM77968
*Scylla serrata*	Anti-lipopolysaccharide factor	ADW11095
*Scylla serrata*	Antilipopolysaccharide factor precursor	ACH87655
*Scylla paramamosain*	Anti-lipopolysaccharide factor	AHB62419
*Scylla paramamosain*	Anti-lipopolysaccharide factor isoform 4	CCW43202
*Scylla paramamosain*	Anti-lipopolysaccharide factor-6	CCW43204
*Scylla tranquebarica*	Anti-lipopolysaccharide factor precursor	AFM77969
*Penaeus monodon*	Anti-lipopolysaccharide factor isoform 3	ABP73289
*Penaeus monodon*	Anti-lipopolysaccharide factor isoform 6	AER45468
*Penaeus monodon*	Anti-lipopolysaccharide factor isoform 2	ABP73291
*Penaeus monodon*	Anti-lipopolysaccharide factor isoform 7	ANP92039
*Litopenaeus vannamei*	Anti-lipopolysaccharide factor isoform 1	AHG99284
*Litopenaeus vannamei*	Anti-lipopolysaccharide factor AV-K isoform	ACT21197
*Litopenaeus vannamei*	Anti-lipopolysaccharide factor 5 (LvALF8)	AVP74305
*Litopenaeus schmitti*	Anti-lipopolysaccharide factor	ABJ90465
*Litopenaeus stylirostris*	Anti-lipopolysaccharide factor	AAY33769
*Macrobrachium rosenbergii*	Anti-lipopolysaccharide factor	AFC93433
*Macrobrachium rosenbergii*	Anti-lipopolysaccharide factor	AEP84102
*Macrobrachium rosenbergii*	Anti-lipopolysaccharide factor 3	AFW04306
*Macrobrachium rosenbergii*	Anti-lipopolysaccharide factor 2	AFW04305
*Macrobrachium rosenbergii*	Anti-lipopolysaccharide factor 1	AFW04304
*Macrobrachium rosenbergii*	Anti-lipopolysaccharide factor 2	ADI80707

**Table 2 marinedrugs-19-00250-t002:** Minimal inhibitory concentration (MIC) and minimal bactericidal concentration (MBC) of the synthetic LPS-binding domain (LBD) peptide.

Microorganisms	MIC	MBC
Gram negative bacteria (G^−^):		
*Vibrio parahemolyticus*	1–2 μM	4–8 μM
*Vibrio harveyi*	1–2 μM	4–8 μM
*Vibrio alginolyticus*	1–2 μM	4–8 μM
*Vibrio owensii*	1–2 μM	4–8 μM
*Photobacterium damselae*	2–4 μM	8–16 μM
*Escherichia coli*	4-8 μM	>64 μM
Gram positive bacteria (G^+^):		
*Staphylococcus epidermidis*	2–4 μM	32–64 μM
*Staphylococcus aureus*	4–8 μM	>64 μM
*Kurthia gibsonii*	4–8 μM	4–8 μM

**Table 3 marinedrugs-19-00250-t003:** Key physicochemical parameters of LvALF8-LBD, FcALF8-LBD and GFP peptides.

Peptide	Sequence	NetC ^a^	H ^b^	pI ^c^	GRAVY ^d^	μH ^e^
**LvALF8-LBD**	Ac-Y(CSYSTRPYFLRWRLKFKSKVWC)P-NH2	6	0.571	10.20	−0.541	0.040
**FcALF8-LBD**	Ac-Y(CSYSTRPYFIRWQLKFKTKIWC)P-NH2	5	0.651	9.90	−0.445	0.080
**GFP**	Ac-TTGKLPVPWPTLVTTFSYGVQCFS-NH2	1	0.732	7.86	0.3333	0.314

^a^ NetC represents net charge. Lys (K), and Arg (R) were assigned with +1 charge; ^b^ H represents average hydrophobicity, that is, total hydrophobicity indices of all residue divided by the number of residues; ^c^ pI represents isoelectric point; ^d^ GRAVY represents the Grand Average hydropathy value; ^e^ μH represents average hydrophobic moment.

## Data Availability

Not applicable.

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
