# Peer review of "A Lymphoid Organ Specific Anti-Lipopolysaccharide Factor from Litopenaeus vannamei Exhibits Strong Antimicrobial Activities"

_marinedrugs, 2021, doi:10.3390/md19050250_

Round 1
Reviewer 1 Report
The study investigates the structure and anti-microbial functions of a newly identified anti-lipopolysaccharide factors (ALF) from the shrimp Litopenaeus vannamei (LvALF8). Authors performed a number of complementary analyses including a phylogenetic analysis of AFLs from 12 crustacean species, the study of tissue distribution of ALF transcripts, various in vitro and in vivo anti-bacterial and anti-viral activity assays and a sequence analysis. The study is straightforward and the results are interesting, specially since they provide new insights into the mechanisms underlying the differences in disease resistance among some shrimp species.
I have several points that may be addressed to improve the manuscripts.
- In the introduction section (line 62-64), authors mentioned that shrimps possess different isoforms of AFL. Isoforms normally refer to splice variants of a same gene, which is not the case here. I would suggest using « ALF » family members, instead of « isoforms »
- In the text as it now stands, it is unclear to me why authors performed the phylogenetic analysis. This study is neither justified in the introduction/objectives, nor commented in the discussion. Please add some text in the relevant sections.
- For the tissue expression study, authors first compared the transcript levels in different tissues and showed that ALF8 is mainly expressed in the lymphoid organ. They then compare ALF expression in this organ only, after infection. It seams possible that expression of ALF may significantly increase in organs other that the lymphoid organ after an infection, and it would have been informative to analyse ALF expression in all organs under standard conditions and after infection. If authors cannot complete this experiment, they should at least comment and mention this possibility.
- For the in vitro antibacterial assays, how does the activity of ALF compare to activity of other ALFs or AMPs? Please comment
- For the in vivo antibacterial assays, authors count bacteria in hepatopancreas. Why this organ? Please add information.
- Finally, although the text is clear and easily understandable, I noticed several errors (e.g., L43 « tolerant » ; L82 « were » ; L102 « evolutionary » , etc etc). Please, go through an additional English editing effort.
Author Response
Thanks for the reviewer's suggestions. The point-to-point responses were uploaded in the attached word file.

Reviewer 2 Report
The manuscript "A Lymphoid Organ Specific Anti-lipopolysaccharide Factor from Litopenaeus vannamei Exhibits Strong Antimicrobial Activities" describes newly identified ALF gene LvALF8 in lymphoid organ of shrimp L. vannamei. LBD peptide of LVALF8 has much higher antimicrobial activity than previously described FcALF8-LBD. Differences in primary sequences of these peptides were also described. This information can not only shed more light into questions about ALF functions in crustaceans but also support design of LBD-inspired antimicrobial drugs.
The article is well designed and clearly written. I would only recommend checking English. I have no other questions or notes.
Author Response
Thanks for the reviewer's suggestion. We have gone through the text carefully and made some revisions in the revised manuscript.